# Evaluating the Roles of Different Types of Laser Therapy in Becker’s Nevus Treatment

**DOI:** 10.3390/jcm11144230

**Published:** 2022-07-21

**Authors:** Muhammad K. Al-Bakaa, Muhsin A. Al-Dhalimi, Prabhatchandra Dube, Fatimah K. Khalaf

**Affiliations:** 1Department of Medicine, University of Jabir ibn Hayyan College of Medicine, Najaf 54006, Iraq; muhammad.k.albakaa@jmu.edu.iq; 2Department of Medicine, University of Kufa College of Medicine, Najaf 54006, Iraq; muhsin.aldhalimi@uokufa.edu.iq; 3Department of Medicine, University of Toledo College of Medicine and Life Sciences, Toledo, OH 43606, USA; prabhatchandra.dube@utoledo.edu; 4Department of Clinical Pharmacy, University of Alkafeel College of Pharmacy, Najaf 54006, Iraq

**Keywords:** Becker’s nevus, Nd: YAG laser, Erbium: YAG laser, split lesion

## Abstract

Becker’s nevus (BN) is a cutaneous hamartoma of benign nature that develops through adolescence and affects mostly young men. The nevus is usually located unilaterally and is characterized by hypertrichosis and hyperpigmentation. Despite recent advances in treatment modalities, no effective treatment has been established for BN hyperpigmentation. We sought to assess the efficacy and safety of fractional Erbium: YAG 2940 nm and Q-switched Nd: YAG 1064 nm lasers in the treatment of BN hyperpigmentation. Twenty-three patients with BN were included in a prospective, randomized-controlled, observer-blinded, split-lesion comparative technique trial. In each patient, two similar square test regions were randomized to either be treated with a fractional Erbium: YAG 2940 nm laser or with a Q-switched Nd: YAG 1064 nm laser. Each patient was treated with three sessions at six-week intervals. At the follow-up, clearance of hyperpigmentation was assessed by physician global assessment, visual analogue scale, grade of improvement, patient global assessment, and patient satisfaction. Regions treated with the fractional Erbium: YAG 2940 nm laser demonstrated significantly better improvement compared to ones treated with the Q-switched Nd: YAG 1064 nm (*p*-value = 0.001) laser. Adverse effects such as repigmentation and hypertrophic scarring were not reported during the follow-up period. The outcomes were cosmetically acceptable with overall high satisfaction among the included patients. Our data suggest a superior role for the fractional Erbium: YAG (2940 nm) laser in the treatment of BN hyperpigmentation compared to the Q-switched Nd: YAG (1064 nm) laser, along with being a safer method and having no reported side effects.

## 1. Introduction

Becker’s nevus (BN), which is also known as Becker’s melanosis, Becker’s pigmentary hamartoma, and pigmented hairy epidermal nevus, is a form of acquired hyperpigmented epidermal nevus [1,2,3]. BN is common in young people with fair skin, with lesions often appearing around puberty [1,4]. BN is typically acquired, though cases with congenital nevus have also been reported [1,5]. The male-to-female ratio varies and ranges from 2:1 to 6:1; hence, it is more common in men and, despite a record of familial cases, the occurrence of BN is mostly sporadic [1,6,7]. Evidence suggests an autosomal dominant mode of inheritance with reduced penetrance, although some researchers have presumed that predominant inheritance better explains the regional predilection and mosaic patterns of BN [6,7,8].

BN classically presents as a large, asymptomatic, hyperpigmented patch with occasional hypertrichosis [9]. After their initial appearance, BN lesions tend to become darker and expand over two years, with a characteristic light tan to dark brown color that can be cosmetically distressing for patients [10]. Various treatment modalities have been examined, but unfortunately, no effective treatment has been proven to successfully treat BN hyperpigmentation [11]. Hence, BN presents a real challenge for physicians. Currently, available treatment options for BN hyperpigmentation, such as laser-assisted pigment removal and topical therapy, have an impermanent effect and can only reduce the pigment stain without removing it completely [11,12,13]. Other treatment options, such as surgical excision with a graft, are usually not recommended because of lesions of large size and the risk of potential scarring [14]. Camouflage makeup is frequently used, especially when conventional therapy is ineffective [15].

Studies that have examined the role of laser treatment in BN have suggested that the hyperpigmentation components of BN are extremely resistant and have a very high recurrence rate, which makes the whole matter even more challenging to both patients and physicians [16]. Due to the rare nature of the condition, earlier studies have a common limitation when it comes to sampling size, an issue that can be highlighted in most BN-related studies. Another limitation is the lack of information regarding the safety of laser treatment in patients with dark skin who have BN; again, this could be due to the infrequent occurrence of the condition. 

On this background, we aim to search for a safe and effective method to treat BN hyperpigmentation. In the current study, we hypothesize that a nonpigment-specific ablative laser (Erbium: YAG 2940 nm laser) has better treatment outcomes in BN hyperpigmentation compared to a pigment-selective laser (Nd: YAG 1064 nm laser), using a split-lesion comparative technique, in which two similar test regions are randomized in each patient to be treated with a different laser and to examine their outcomes side-to-side. This method helps us maximize our sample size and eliminate interindividual variability. The current study also examines BN laser treatment in patients with dark skin. BN is a major cosmetic issue in patients with darker skin phenotypes, especially females, with no studies reporting a safe and effective method of laser treatment in those rare, yet very important, populations. Thus, our study aims to assess and compare the efficacy and safety of fractional Erbium: YAG 2940 nm and Q-switched Nd: YAG 1064 nm lasers in the treatment of BN hyperpigmentation in a population with dark skin tones.

## 2. Methods

### 2.1. Patients

The prospective interventional therapeutic clinical study was performed at the Center of Lasers and Research, College of Medicine, University of Kufa, during the period from October 2017 to December 2018. Thirty-two patients were enrolled in this trial. The included patients were in good general health and presented with typical clinical features of Becker’s nevus (sharply demarcated, unilateral, hyperpigmented patch with or without hypertrichosis and onset during the peripubertal period). Patients with the following conditions were excluded from the study: presence of active skin infection in the lesional area; pregnant and lactating women; patients with chronic dermatological conditions (such as eczema, psoriasis, etc.); patients who used topical depigmenting therapy within the previous 6 months or used any laser therapy to treat BN within the previous year; recent high exposure to sunlight or ultraviolet light; personal or family history of poor wound healing; presence of a hypertrophic scar and keloid; or prior therapy with isotretinoin in the last 6 months.

This study was approved by the ethical committee of the Arab Academic Board of Medical Specialization in Dermatology and Venereology. The procedure was fully explained, and consents were signed by the patients after understanding the nature of the study, the anticipated benefits, and the potential risks. High-quality colored photographs were captured for all the participating patients to record the lesion area at baseline, at the end of the last session, and at the follow-up using a high-sensitivity Sony digital 24 megapixel D5200 AF-S DX Zoom-Nikkor camera. Camera settings, distance, and illumination were considered to eliminate possible variability.

### 2.2. Treatment Protocol

Each lesion was divided longitudinally into two parts, and every part was treated randomly with either a fractional Erbium: YAG 2940 nm laser (Quanta System—DNA laser technology—MATISSE, Milan, Italy) or a Q-switched Nd: YAG 1064 nm laser (Quanta system—DNA laser technology—ULTRALIGHT, Milan, Italy). Patients were treated for three sessions with a six-week interval between each session. Parameters for the laser devices were selected according to the manufacturers’ recommendations and were fixed for all patients. For each laser type, the parameters were selected to equivalently penetrate the pigment depth. The parameters for the fractional Erbium: YAG laser were: a wavelength of 2940 nm, a fluence of 10 J/cm^2^, a spot size of 9 mm, a pulse duration of 0.35 ms, and a frequency of 6 Hz. Whitish, desiccated skin was used as the clinical endpoint during laser treatment. The Q-switched Nd: YAG laser parameters were: a wavelength of 1064 nm, a fluence of 10 J/cm^2^, a spot size of 3 mm, a pulse duration of 10 ns, and a frequency of 10 Hz. Uniform, immediate whitening without epidermal disruption was used as the clinical endpoint during laser treatment; however, for facial lesions, a fluence 5 J/cm^2^ was used on the part treated with the Q-switched Nd: YAG laser, as the clinical endpoint was crossed when using 10 J/cm^2^ with macroscopic pinpoint bleeding. Some areas of superficial erosion ended with postinflammatory hyperpigmentation, which usually happened when the case was first treated with a fluence 10 J/cm^2^. Before each laser therapy, patients who had associated hypertrichosis were instructed to remove the terminal hair either by shaving or using a topical hair removal cream (Fem^®^ USA hair removal cream) one day before the treatment session. Topical anesthetic cream (lidocaine 10.56% cream) was applied, and the lesion area was then covered for 40 min before the procedure. During the procedure, a cold air chiller device (Zimmer MedizinSysteme, Neu-Ulm, Germany) was used as a precooling technique to minimize epidermal damage. Patients were asked to lie down in either a supine or prone position according to the BN site. Patients’ eyes were covered with protective goggles. The handpiece was held perpendicular to the target tissue when the parts were treated with the fractional Erbium: YAG 2940 nm laser. Two passes of pulses were delivered, scanning the lesional area in two directions (horizontally and vertically) with 10% overlapping. Whitish, desiccated skin was eliminated with saline-soaked gauze after the first pass but left after the second pass as a protective barrier. An extra pass was done at the periphery of the treated area to provide natural blending and minimize the demarcation of the surrounding untreated skin. One pass mode with 10–20% overlapping was used for the part treated with the Q-switched Nd: YAG 1064 nm laser. The optimal clinical endpoint during laser treatment was immediate whitening of the lesion without epidermal disruption. During the treatment session, cold air was applied to minimize epidermal damage and pain. At the end of the session, treated sites were observed, and early skin reactions to laser treatment were recorded. All patients were asked to report whether the procedure was painful and, if so, the degree of pain or discomfort they felt during laser treatment was recorded. 

The patients were strictly instructed to avoid excessive sun exposure and to use a broad-spectrum sunscreen with a sun protection factor (SPF) of 30 or more in the morning and reapply it every 2 h during the treatment period and at follow-up intervals. Patients were given a topical mixture of 1% hydrocortisone and fucidic acid ointment to apply two times daily for five days and were instructed not to use any type of therapy during the study and follow-up period. Patients were evaluated every six weeks throughout the treatment period regarding signs of improvement and to check for any possible complications. 

### 2.3. Randomization

A simple method of randomization was used in which the patients randomly selected a paper containing the type of laser that would be used for each part of their BN lesion. 

### 2.4. Efficacy Evaluation

Treated patients were assessed both objectively and subjectively to evaluate the effects of the treatments using the following methods: 

A: Objective methods:

The photographs were blindly assessed at the end of the study by two independent expert dermatologists, and their evaluations regarding the degree of hyperpigmentation and the grade of improvement for each half of the lesion were recorded. The average improvement of both sides was taken for each patient. The following methods were used:The visual analogue scale (VAS), scored from 0 to 10, represented increasing levels of lesional hyperpigmentation and was used to assess the level of lesional pigmentation and monitor the improvement following therapy. The grade of hyperpigmentation for each part of the lesion was recorded at the baseline, at the end of the last session, and for the follow-up period. A score of (0) indicated that lesion color was close to the color of the surrounding nonlesional skin, while a score of (10) meant the lesion color was extremely dark brown.The degree of improvement was evaluated using a six-point scale, where failure = 0% improvement, mild = 1–25% improvement, moderate = 26–50% improvement, good = 51–75% improvement, excellent = 76–99% improvement, and perfect = 100% improvementThe degree of repigmentation was examined using a five-point scale: Grade 1 (no repigmentation), Grade 2 (mild (1–39%) repigmentation), Grade 3 (moderate (40–69%) repigmentation), Grade 4 (severe (70–100%) repigmentation), and Grade 5 (worsening of hyperpigmentation)

B: Subjective methods:

This method included recording the degree of patient satisfaction regarding the improvement of hyperpigmentation at the end of the last session and at the follow-up period for each half of the lesion. The degree of satisfaction was graded from 0 to 10, in which a grade of (0) meant the patient was completely unsatisfied, while a grade of (10) indicated that the patient was highly satisfied by the results.

Statistical Analysis:

All statistical analyses were performed using GraphPad PRISM 7 software (San Diego, CA, USA). Data are presented as the mean ± standard error of the mean. Student’s unpaired *T*-test was used to assess statistically significant differences between the two groups. One-way ANOVA and post hoc multiple comparison tests were used when comparing more than two groups. Statistical significance was accepted as *p* < 0.05.

## 3. Results

Twenty-three out of thirty-two patients completed the treatment sessions. The other nine patients were excluded because they did not follow the treatment protocol strictly. Table 1 and Table 2 summarize the sociodemographic and lesional characteristics of the studied patients. 

### 3.1. Grade of Improvement 

Areas treated with the fractional Erbium: YAG (2940 nm) laser reported a significant grade of improvement after treatment compared to the Q-switched Nd: YAG-treated areas. Our data showed a certain grade of improvement in 100% of the treated patients compared to only 65.22% of patients treated with the Q-switched Nd: YAG laser (Table 3). While the Q-switched Nd: YAG laser demonstrated a 35% failure in treatment, the fractional Er: YAG laser reported 0% failure.

### 3.2. Degree of Improvement

The degree of improvement based on the visual analogue scores of the first and second assessor showed a highly considerable difference between parts of the BN treated with the fractional Erbium: YAG laser compared to the parts treated with the Q-switched Nd: YAG laser (*p*-value < 0.0001) (Figure 1A,B).

### 3.3. Efficacy of Treatment

Similarly, the fractional Erbium: YAG laser demonstrated a significantly improved treatment efficacy with better outcomes in the parts treated with the fractional Erbium: YAG (2940 nm) laser compared to the Q-switched Nd: YAG laser (1064 nm) with *p*-value < 0.003 and *p*-value < 0.001, respectively (Figure 2A,B). 

### 3.4. Patient Satisfaction

Parts treated with the fractional Erbium: YAG (2940 nm) laser reported significantly higher satisfaction scores by patients compared to the parts treated with the Q-switched Nd: YAG (1064 nm) laser (Figure 3). Representative images of a patient before and after treatment (Figure 4) shows that the pigmented lesion split into two areas, one assigned to the fractional Erbium: YAG laser and the other one assigned to the Q-switched Nd: YAG (1064 nm) laser before and after treatment.

### 3.5. Side Effects

All patients developed a slight tingling sensation at the time of treatment and mild erythema, which resolved within a few hours on the side treated with the fractional Erbium: YAG laser; however, on the side treated with the Q-switched Nd: YAG laser, patients developed a burning sensation and transient erythema that remained for no more than 2 days after each treatment session. One patient (4.37%) with facial BN developed mild postinflammatory hyperpigmentation on the side that was treated with the Q-switched Nd: YAG laser, which significantly improved after the application of a topical combination of 1% hydrocortisone, 0.025% tretinoin, and 2% hydroquinone cream once at night for 1 month. This occurred when the first case with facial BN was treated using a fluence of 10 J/cm^2^ in the first session. No other significant side effects were recorded.

## 4. Discussion

Becker nevus (BN) remains a therapeutically challenging condition despite various currently available technologies [17]. To date, no specific treatment has proved to be effective for the hyperpigmented component of BN [18]. Studies investigating the hyperpigmentation component of BN have reported the condition to be very resistant to treatment, and even though laser therapy could slightly reduce the pigment, enormous inter-individual variability in responses, along with high recurrence of pigmentation, have been reported [16].

The present work aimed to evaluate and compare the efficacy of fractional Erbium: YAG (2940 nm) and Q-switched Nd: YAG (1064 nm) lasers in the treatment of BN. Using a split-lesion technique in which both laser types were used side-by-side in patients to eliminate interindividual variability and to ensure a fair comparison, this technique provided a bigger sample size and higher statistical power when compared with previous studies examining BN. While a nonfractional Erbium: YAG 2940 nm laser was previously examined for treating BN hyperpigmentation, our study is the first to assess a fractional Erbium: YAG 2940 nm laser in the treatment of BN hyperpigmentation [16].

The Erbium: YAG laser treats the hyperpigmented components of BN through tissue ablation [19], with water being the target chromophore. The Erbium: YAG laser removes the superficial pigmented lesions via thermal damage that leads to the destruction of the lesion by denuding the epidermis [20]. Fractional Erbium: YAG causes small thermal injury zones known as microthermal treatment zones (MTZs) that create microscopic vaporization channels that can extend up to 3–4 mm into the dermis (Figure 5). Basal layer pigment, along with a small amount of papillary dermal debris, is removed with the desquamation of the epidermal portion of each MTZ. The depth of these MTZs can be determined by the operator by adjusting the settings. Because spared tissue between the columns of ablation initiates remodeling, the depth of the ablation can be extended far deeper than with traditional ablative resurfacing procedures [21]. 

The Q-switched Nd: YAG laser targets the endogenous melanin pigment to treat BN hyperpigmentation. The Q-switched Nd: YAG laser is considered a highly pigment-selective laser [22,23], and it selectively destructs epidermal or dermal melanin, sparing the epidermis [24]. It emits pico- or nanosecond pulses at a wavelength that is absorbed more avidly by melanin. Hence, the energy targets the pigment directly without causing any alterations in the surrounding tissues (Figure 5). The Q-switched Nd: YAG 1064 nm laser was shown to destroy both melanocyte and dermal melanophages [25]. It was demonstrated to be useful in the treatment of pigmented lesions with both dermal and epidermal components, such as in BN. It also has a superior role in treating dark-pigmented (Fitzpatrick skin types (IV-VI), as it reduces the risk of epidermal injury [26,27]. Landthaler et al. and Anderson et al. showed that multiple treatment sessions were needed to obtain acceptable clearance of hyperpigmentation when using a Q-switched Nd: YAG laser, with complete elimination of the pigment only achieved by continuous, regular sessions of high-fluence energy. Therefore, patients require infinite sessions of Q-switched Nd: YAG treatment to maintain satisfying results. In addition, repigmentation has been observed in some cases after the last treatment session [28,29]. All these findings make the Q-switched Nd: YAG laser unfavorable in treating BN hyperpigmentation. 

Trelles et al. performed a comparative study between the nonfractional ablative Er: YAG (2940 nm) and the Q-switched Nd: YAG (1064 nm) lasers in 22 patients. The patients were divided into two groups: 11 patients were treated with the Er: YAG laser, and 11 patients were treated with the Q-switched Nd: YAG 1064 nm laser. The patients were followed-up with after two years. Complete clearance was observed in six patients (54%) treated with the Er: YAG laser, and moderated clearance (more than 50%) was seen in the remaining patients. In contrast, the Q-switched Nd: YAG-treated group only showed one patient with exhibited marked clearance, with moderate clearance (26–50% reduction) in 45.5% of the patients and mild clearance (1–25% reduction) in 27.3% of the patients. Additionally, treatment effects were reported to last longer, with mild repigmentation in the group treated with the Er: YAG laser compared to the Nd: YAG laser [30,31]. In our study, the results using the Q-switched Nd: YAG (1064 nm) laser were comparable to previous studies but with fewer side effects. This might be due to the optimum cooling procedure used. 

In a pilot study by Al-Saif et al. in which he used a nonfractional ablative Er: YAG 2940 nm laser in BN treatment, a split-lesion treatment was performed, with one part treated with a nonfractional ablative Er: YAG 2940 nm laser, while the other parts remained without treatment (served as a control). The treatment was performed in one session with three to five passes, and only seven patients completed the study to the end. After a 12-month follow-up, a considerable reduction in hyperpigmentation was observed, with five patients showing good improvement (51–75% improvement) and two showing moderate improvement (26–50% improvement). Erythema and postinflammatory hypopigmentation were noticed in all the patients [32]. Interestingly, postinflammatory hypopigmentation was not observed in the current study using the fractional Erbium: YAG (2940 nm) laser. This opens up a potential therapeutic target for BN hyperpigmentation, especially in patients with dark skin tones. In our study, the fractional Erbium: YAG (2940 nm) laser showed significantly better results compared to the Q-switched Nd: YAG (1064 nm) laser, with no adverse side effects, such as scar development and hypopigmentation, compared to other reported methods. The patients were also more pleased with the clinical outcomes of the area treated with the fractional Erbium: YAG (2940 nm) laser treatment compared to the area treated with the Q-switched Nd: YAG (1064 nm) treatment. Hence, our data suggested that the fractional Erbium: YAG (2940 nm) laser was more effective in the treatment of hyperpigmented components of Becker’s nevus compared to the Q-switched Nd: YAG (1064 nm) laser. 

## Figures and Tables

**Figure 1 jcm-11-04230-f001:**
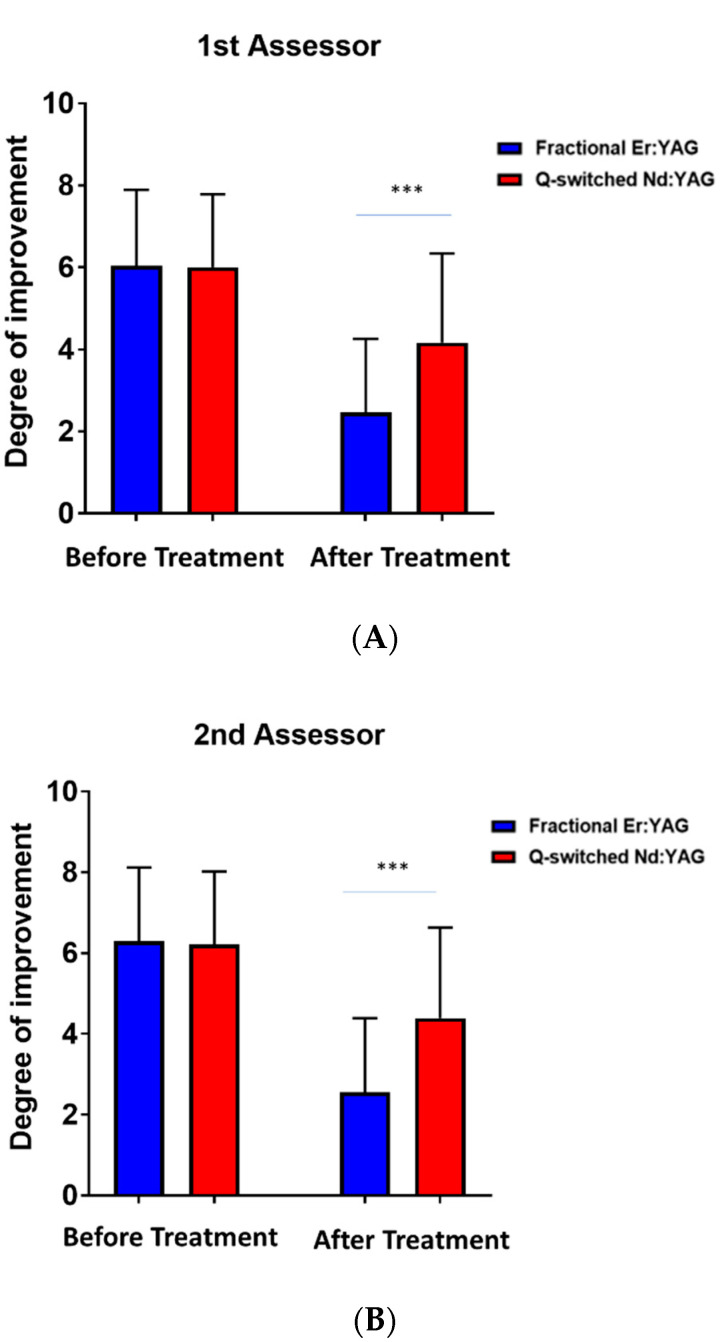
The degree of improvement based on the visual analogue scores of (**A**) the first assessor before and after treatment between the Q-switched Nd: YAG (1064 nm) and fractional Er: YAG (2940 nm) lasers and (**B**) the second assessor before and after treatment between the Q-switched Nd: YAG (1064 nm) and fractional Er: YAG (2940 nm) lasers. *** *p* < 0.001.

**Figure 2 jcm-11-04230-f002:**
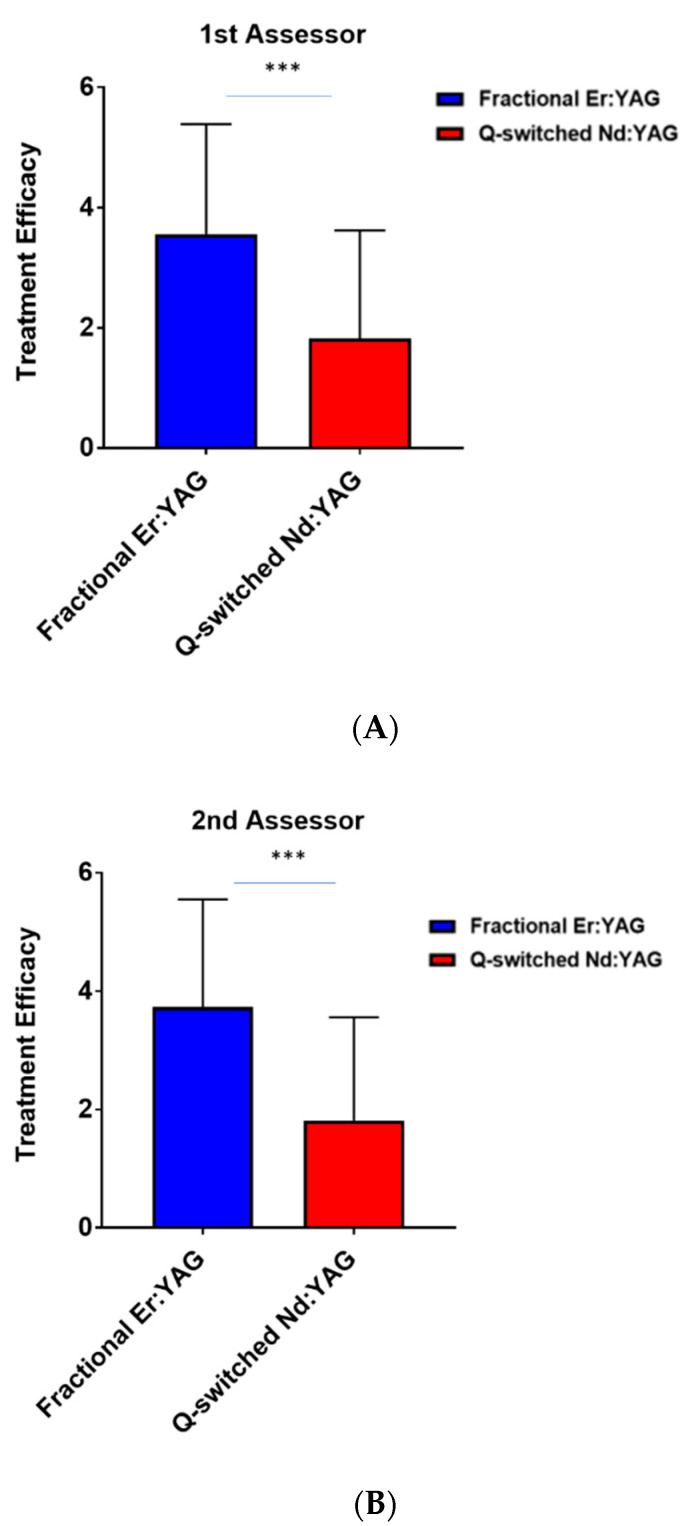
Treatment efficacy based on visual analogue scores of (**A**) the first assessor between Q-switched Nd: YAG (1064 nm) and fractional Er: YAG (2940 nm) lasers and (**B**) the second assessor between Q-switched Nd: YAG (1064 nm) and fractional Er: YAG (2940 nm) lasers. *** *p* < 0.001.

**Figure 3 jcm-11-04230-f003:**
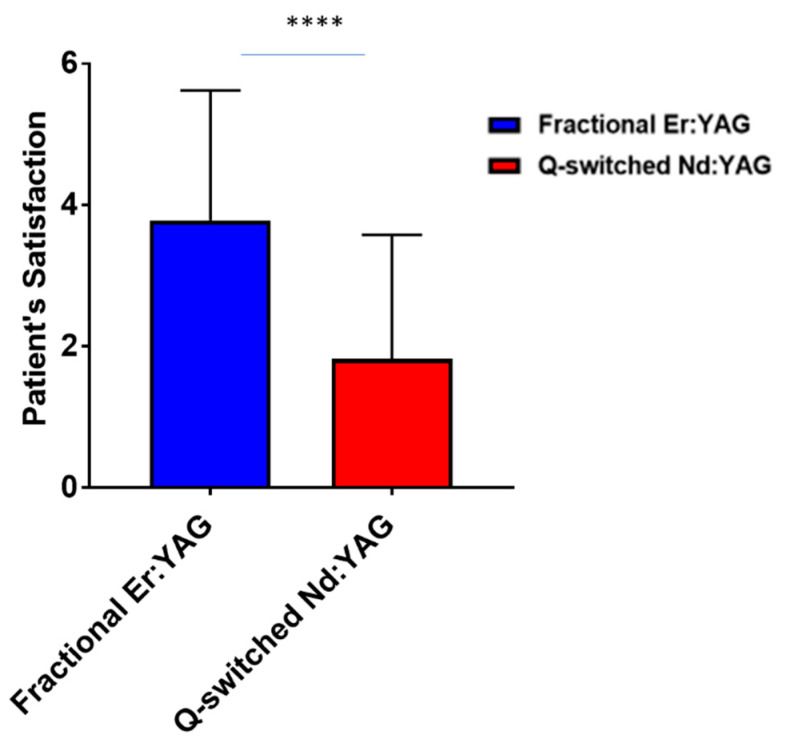
Patient satisfaction based on the treatment outcomes between Q-switched Nd: YAG (1064 nm) and fractional Er: YAG (2940 nm) lasers. **** *p* < 0.0001.

**Figure 4 jcm-11-04230-f004:**
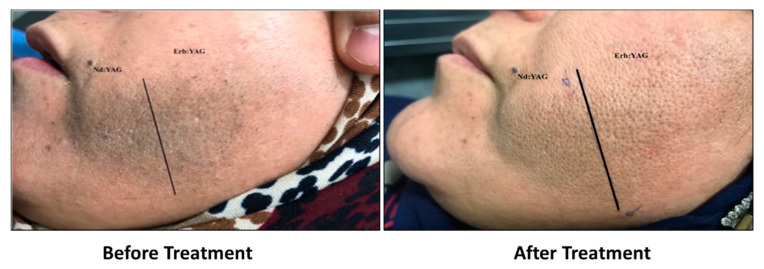
Forty-three-year-old female with facial Becker’s nevus before treatment and 2 weeks after the last treatment session with lesion area divided into two parts: one treated with fractional Erbium: YAG 2940 nm laser and the other part treated with Q-switched Nd: YAG 1064 nm laser.

**Figure 5 jcm-11-04230-f005:**
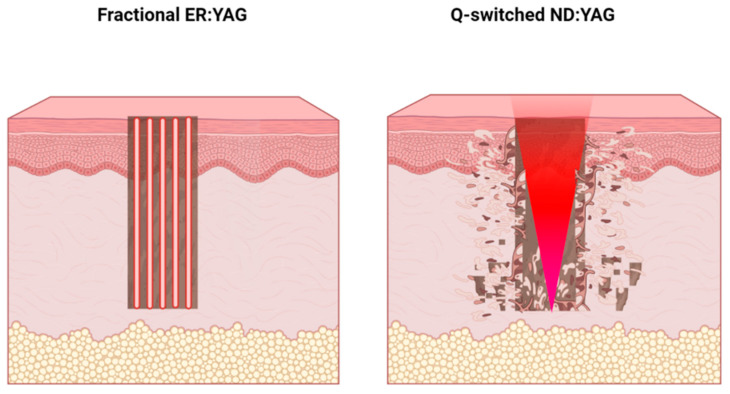
Schematic illustrating the laser–tissue interaction for the fractional Erbium: YAG 2940 nm laser (**left**) and Q-switched Nd: YAG 1064 nm lasers (**right**).

**Table 1 jcm-11-04230-t001:** Sociodemographic characteristics of the studied patients.

Patient Characteristics		Range	Mean
Age/years		10–58	23.65 ± 10.86
Age of onset/years		7–14	10.86 ± 2.04
		Number	Percent
Age group/years	<20	9	39.13%
	20–40	12	52.17%
	>40	2	8.69%
Gender	Male	8	34.78%
	Female	15	65.21%
Residency	Urban	21	91.30%
	Rural	2	8.69%
Skin types	Type III	10	43.47%
	Type IV	13	56.5%

**Table 2 jcm-11-04230-t002:** Lesion characteristics of the studied patients.

Lesion Characteristics		Number	Percent
Lesion localization	Face	3	13.04%
	Neck	1	4.37%
	Upper trunk	6	26.08%
	Upper extremity	3	13.04%
	Lower extremity	1	4.37%
	Upper trunk with upper extremity	9	39.13%
Lesion color	Light brown	3	13.04%
	Moderate brown	11	47.82%
	Dark brown	7	30.43%
	Extremely dark brown	2	8.69%
Hypertrichosis	Positive	6	26.08%
	Negative	17	73.91%
Hair density	Mild	4	17.39%
	Moderate	1	4.37%
	Marked	1	4.37%
Associated acneiform lesions	Positive	2	8.69%
	Negative	21	91.30%
Previous treatment	Laser (before 1 year)	5	21.73%
	Topical therapy (before 6 months)	2	8.69%
	None	16	69.56%

**Table 3 jcm-11-04230-t003:** The grade of improvement between Q-switched Nd: YAG (1064 nm) and fractional Er: YAG (2940 nm) lasers after treatment.

Grade of Improvement	After Treatment		*p* Value
	Fractional Er:YAG (2940 nm) No. (%)	Q-Switched Nd:YAG (1064 nm) No. (%)	
Failure (0%)	0 (0%)	8 (34.78%)	0.001
Mild (1–25%)	5 (21.79%)	4 (17.39%)	
Moderate (26–50%)	3 (13.04%)	6 (26.08%)	
Good (51–75%)	8 (34.78%)	2 (8.69%)	
Excellent (76–99%)	3 (13.04%)	3 (13.04%)	
Perfect (100%)	4 (17.39%)	0 (0%)	

## Data Availability

The datasets generated and analyzed during the current study are available from the corresponding author on reasonable request.

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
