# Peer review of "Evaluating the Roles of Different Types of Laser Therapy in Becker’s Nevus Treatment"

_jcm, 2022, doi:10.3390/jcm11144230_

Round 1

Reviewer 1 Report

This manuscript aims to search for a safe and effective method to treat BN hyperpigmentation. The authors hypothesize that a non-pigment-specific ablative laser (Erbium: YAG 2940 nm laser) has better treatment outcomes in BN hyperpigmentation compared to a pigment-selective laser (Nd:YAG 1064nm laser). An ingenious trial is designed using split lesion comparative technique, in which two similar test regions were randomized in each patient to be treated with a different laser and examine their outcomes side to side. I think this work can be considered to be published, with following comments,

 1) line 67-69, “we hypothesize that a non-pigment-specific ablative laser has better treatment outcomes in BN hyperpigmentation compared to a pigment-selective laser”

Is suggested to revised as

we hypothesize that a non-pigment-specific ablative laser (Erbium: YAG 2940 nm laser) has better treatment outcomes in BN hyperpigmentation compared to a pigment-selective laser (Nd:YAG 1064nm laser).

 2) The clinical study was performed during the period from October 2017 to December 2018. Did the authors conduct follow-up assessment of the therapy efficacy after 4 years?

 3) Line 137, this sentence about cooling is redundant, which has been stated in line 124.

 4) The Male to female ratio varies and ranges from 2:1 to 6:1. But among the 23 patients, male to female is 8:15?

 5) During the comparison, the parameters of the two lasers are quite different except fluence. Spot sizes are 9 mm and 3mm, pulse durations are 0.35 ms and 10 ns, while frequencies are 6 Hz and 10 Hz for 2940 and 1064 nm lasers, respectively. This is worth to be discussed more detailed.

 6) For hyperpigmentation, the 585/595nm pulsed dye laser is more highly pigment-selective than Nd:YAG laser. The authors may compare the wavelength selection in the 4th paragraph in the discussion part.

Author Response

Responses to the Comments

We are grateful to the reviewers and editor for the thoughtful comments and insightful suggestions that helped us improve our manuscript considerably. As indicated in the responses below, we have taken all their valuable comments and suggestions into consideration in the revised manuscript. Please, note that the Reviewers’ comments are written in black, and our responses are written in blue.

Reviewer 1

This manuscript aims to search for a safe and effective method to treat BN hyperpigmentation. The authors hypothesize that a non-pigment-specific ablative laser (Erbium: YAG 2940 nm laser) has better treatment outcomes in BN hyperpigmentation compared to a pigment-selective laser (Nd:YAG 1064nm laser). An ingenious trial is designed using split lesion comparative technique, in which two similar test regions were randomized in each patient to be treated with a different laser and examine their outcomes side to side. I think this work can be considered to be published, with following comments,

 1) line 67-69, “we hypothesize that a non-pigment-specific ablative laser has better treatment outcomes in BN hyperpigmentation compared to a pigment-selective laser”

Is suggested to revised as

“we hypothesize that a non-pigment-specific ablative laser (Erbium: YAG 2940 nm laser) has better treatment outcomes in BN hyperpigmentation compared to a pigment-selective laser (Nd:YAG 1064nm laser).”

We thank the Reviewer for this important suggestion and in response, we have revised the sentence as per the reviewer's suggestion. Please refer to the revised manuscript on page 2 (yellow highlighted).

 2) The clinical study was performed during the period from October 2017 to December 2018. Did the authors conduct a follow-up assessment of the therapy efficacy after 4 years?

We thank the Reviewer for highlighting this important consideration. In fact, we are currently working on a follow-up study of the therapy efficacy. We are compiling data from 1,2-, and 4-years' time points. Hopefully, will have a continuation study to examine this aspect.

 3) Line 137, this sentence about cooling is redundant, which has been stated in line 124.

We thank the Reviewer for this valuable comment and in response, we have deleted this sentence to avoid repetition. Please refer to the revised manuscript on page 3.

 4) The Male to female ratio varies and ranges from 2:1 to 6:1. But among the 23 patients, male to female is 8:15?

We thank the Reviewer for this very important point. We speculate that, though men are affected by BN more than females. It appears like females tend to seek medical advice to remove the BN pigment more than males do. We think this might be due to cosmetic reasons. It is a very interesting observation that is worth investigating. We appreciate the reviewer for pointing out this point. In the future, will conduct a survey study to address this very important point.

 5) During the comparison, the parameters of the two lasers are quite different except for fluence. Spot sizes are 9 mm and 3mm, pulse durations are 0.35 ms and 10 ns, while frequencies are 6 Hz and 10 Hz for 2940 and 1064 nm lasers, respectively. This is worth to be discussed more detail.

We thank the Reviewer for highlighting this important consideration and in response. The parameters were selected based on the target depth. BN pigment has epidermal and dermal components, hence for each laser type, we have selected the parameters that equivalently penetrate the pigment depth. We have now added a statement to clarify this matter. Please refer to the revised manuscript on page 3 (yellow highlighted).

 6) For hyperpigmentation, the 585/595nm pulsed dye laser is more highly pigment-selective than Nd:YAG laser. The authors may compare the wavelength selection in the 4th paragraph in the discussion part.

We thank the Reviewer for this valuable comment and agree that it is important to address this aspect. Though 585/595nm pulsed dye laser is highly pigment-selective it is useful in pigment with epidermal components only. Since BN pigment has epidermal and dermal components, Nd: YAG 1064 nm laser could cover both epidermal and dermal components. We have addressed the selection in the 4th paragraph in the discussion section. Please refer to the revised manuscript on page 11(yellow highlighted).

Reviewer 2 Report

No effective treatment protocol has been established for Becker's nevus. Each comparative study between the results of two types of lasers provides details on which practitioners and other studies will be based. The interesting topic of the paper falls within the scope of JCM. The results of the study are important with high scientific soundness. The experimental and analysis methodology is suitable with high feasibility. There are sufficient details given to replicate the proposed analysis. The discussion is very relevant and complete.

Author Response

Responses to the Comments

We are grateful to the reviewers and editor for the thoughtful comments and insightful suggestions that helped us improve our manuscript considerably. As indicated in the responses below, we have taken all their valuable comments and suggestions into consideration in the revised manuscript. Please, note that the Reviewers’ comments are written in black, and our responses are written in blue.

Reviewer 2

No effective treatment protocol has been established for Becker's nevus. Each comparative study between the results of two types of lasers provides details on which practitioners and other studies will be based. The interesting topic of the paper falls within the scope of JCM. The results of the study are important with high scientific soundness. The experimental and analysis methodology is suitable with high feasibility. There are sufficient details given to replicate the proposed analysis. The discussion is very relevant and complete.

We truly appreciate and thank the Reviewer for this valuable feedback.